# Airway Care Interventions for Invasively Ventilated Critically Ill Adults—A Dutch National Survey

**DOI:** 10.3390/jcm10153381

**Published:** 2021-07-30

**Authors:** Willemke Stilma, Sophia M. van der Hoeven, Wilma J. M. Scholte op Reimer, Marcus J. Schultz, Louise Rose, Frederique Paulus

**Affiliations:** 1Center of Expertise Urban Vitality, Faculty of Health, Amsterdam University of Applied Science, 1105 BD Amsterdam, The Netherlands; f.paulus@amsterdamumc.nl; 2Department of Intensive Care Medicine, Amsterdam University Medical Centers, Location AMC, 1105 AZ Amsterdam, The Netherlands; sophiavanderhoeven@hotmail.com (S.M.v.d.H.); marcus.j.schultz@gmail.com (M.J.S.); 3Department of Cardiology, Amsterdam University Medical Centers, Location AMC, 1105 AZ Amsterdam, The Netherlands; wilma.scholteopreimer@hu.nl; 4Laboratory of Experimental Intensive Care and Anesthesiology (L·E·I·C·A), Location AMC, 1105 AZ Amsterdam, The Netherlands; 5Mahidol—Oxford Tropical Medicine Research Unit (MORU), Mahidol University, Bangkok 10400, Thailand; 6Nuffield Department of Medicine, University of Oxford, Oxford OX3 7BN, UK; 7Florence Nightingale Faculty of Nursing, Midwifery and Palliative Care, King’s College London, London WC2R 2LS, UK; louise.rose@kcl.ac.uk

**Keywords:** intensive care, invasive ventilation, heated humidification, nebulization therapy, manual hyperinflation, mechanical in-exsufflation

## Abstract

Airway care interventions may prevent accumulation of airway secretions and promote their evacuation, but evidence is scarce. Interventions include heated humidification, nebulization of mucolytics and/or bronchodilators, manual hyperinflation and use of mechanical insufflation-exsufflation (MI-E). Our aim is to identify current airway care practices for invasively ventilated patients in intensive care units (ICU) in the Netherlands. A self–administered web-based survey was sent to a single pre–appointed representative of all ICUs in the Netherlands. Response rate was 85% (72 ICUs). We found substantial heterogeneity in the intensity and combinations of airway care interventions used. Most (81%) ICUs reported using heated humidification as a routine prophylactic intervention. All (100%) responding ICUs used nebulized mucolytics and/or bronchodilators; however, only 43% ICUs reported nebulization as a routine prophylactic intervention. Most (81%) ICUs used manual hyperinflation, although only initiated with a clinical indication like difficult oxygenation. Few (22%) ICUs used MI-E for invasively ventilated patients. Use was always based on the indication of insufficient cough strength or as a continuation of home use. In the Netherlands, use of routine prophylactic airway care interventions is common despite evidence of no benefit. There is an urgent need for evidence of the benefit of these interventions to inform evidence-based guidelines.

## 1. Introduction

Critically ill patients receiving invasive ventilation are at risk for retention of airway secretions [1]. The relatively dry gases used during invasive ventilation cause mucosa in the airways to produce more mucus. Moreover, the presence of the endotracheal tube hampers mucociliary clearance [1,2]. Critically ill patients frequently have an impaired cough reflex due to depressed levels of consciousness, sedation, or muscle weakness. For these reasons, intensive care nurses apply interventions that help with evacuation of airway secretions in patients receiving invasive ventilation. 

Within the domain of intensive care nursing, several interventions aiming at prevention of airway secretion accumulation or promotion of airway secretion evacuation have become part of daily care for critically ill invasively ventilated patients. Active humidification i.e., use of heated humidification, helps to prevent production and thickening of airway secretions [3]. Nebulization of mucoactive agents is thought to reduce accumulation of thick and sticky airway secretions [4]. Mucoactive agents are often used in combination with bronchodilators to enhance mobilization of mucus by opening the airways [5,6,7]. Manual hyperinflation [8,9,10] or mechanical in–exsufflation (MI–E), (commonly referred to as cough assist) [8,11,12], may be helpful techniques in mobilizing airway secretions from smaller to larger airways, where it can be removed using suctioning. In the Netherlands, these interventions are mainly performed by intensive care nurses, while involvement by physiotherapists is rare; they focus more on the traditional rehabilitation procedures. Despite common, and in some cases daily use of these airway care interventions, there is a remarkable lack of evidence for clinical benefit [10,12,13,14]. Current practice guidelines [15,16] are primarily based on expert opinion. This lack of evidence may lead to variable use of airway care interventions in daily practice based on local preferences. Our objective was to determine current airway care practices within the domain of intensive care nursing for (1) heated humidification; (2) nebulization therapy; (3) manual hyperinflation; and (4) MI-E in adult intensive care units (ICUs) in the Netherlands. A secondary objective was to investigate perceptions of safety, necessity, and efficacy of these interventions. Our hypothesis was that current practice of, and perceptions towards airway care interventions would be highly variable due to the lack of evidence.

## 2. Materials and Methods

### 2.1. Study Design 

We used a self–report web–based cross-sectional survey design. 

### 2.2. Survey Development and Formatting

The research team, with extensive experience in invasive ventilation and airway care for critically ill patients, iteratively developed the survey. We generated potential items by searching for relevant studies in the MEDLINE and Cochrane databases. During the selection of items we focused on interventions within the domain of intensive care nursing. We did not include chest physiotherapy, like rib cage compression and other techniques aimed at flow augmentation. Previous surveys on this topic were also used to generate items [9,17]. Item reduction occurred through discussion among the research team.

This survey comprised of 58 items, four related to ICU demographics and the remainder grouped within the airway care interventions of interest. We used skip logic when appropriate to enable provision of questions based on participant responses to preceding questions. Questions comprised intensity of use, indications and contraindications, specifics on how the intervention was applied, and how ICU team members were trained in the interventions. For MI-E, we asked additional questions on years of experience with its use in their ICU, who would prescribe and/or apply MI-E, and barriers to use. Intensity of use consisted of three categories. (1) “Routine”, defined as an intervention used prophylactically in all invasively ventilated patients and ordered with a set frequency per day; (2) “on indication”, defined as initiated based on individual patient clinical characteristics; (3) “never”, defined as never used in their ICU.

Perceptions on the safety, necessity, or efficacy of airway interventions were assessed using six statements with a visual analogue scale ranging from 0 to 100 mm.

### 2.3. Survey Pilot Testing

The survey was loaded on to SurveyMonkeyTM [18] and was pilot tested by four ICU nurses and one intensivist from 3 different hospitals [19]. All four had experience in ICU for more than 5 years and were currently working clinically. Every pilot tester returned a checklist after testing with questions on face and construct validity including clarity, redundancy, and completeness of items; suggestions for additional items required time to complete. After pilot testing minor revisions were made, skip logic was corrected, and pictures of nebulizer types were added for clarity. 

### 2.4. Sample 

Our sample comprised all adult ICUs in the Netherlands. We contacted each ICU by telephone in November 2017 to identify one senior healthcare professional who would take responsibility for survey completion on behalf of their ICU. This person was responsible for invasive ventilation policy and procedures, and either an ICU nurse, advanced ventilation nurse specialist, or physician. Advanced ventilation nurse specialists complete an additional education 14 month program on mechanical ventilation 240 study hours: 1.4 European Credit Transfer and Accumulation System (ECTS). 

### 2.5. Survey Administration

We sent an email with instructions and the secure survey link to participants on March 2018, with 3 survey completion reminders sent over 6 weeks. Survey instructions explicitly stated the respondent was to report on current practices in their ICU (i.e., not their personal preferences). For statements regarding perceptions of the efficacy and safety of airway care interventions, we instructed respondents to provide their personal view.

### 2.6. Analysis Plan

Frequencies and proportions were used to describe categorical data. Proportions were reported as percentages. A heat map was constructed to visualize the variability in practice of airway care interventions [20]. Airway care interventions and intensity of their use were displayed. Perceptions of respondents on statements were visualized in boxplots with means and interquartile ranges. A score of 50 was used as a threshold for agreement or disagreement. In a posthoc analysis, differences in use of airway care interventions were compared between academic-teaching hospitals and general hospitals, as well as in ICUs with ≤20 beds compared to >20 beds. In addition, associations of hospital type or ICU size on the use of airway care interventions were explored by separate logistic regression models (e-supplement). Analyses were performed using SPSS (IBM SPSS Statistics 25) and R language and environment for statistical computing [21].

### 2.7. Ethical Considerations

The Institutional Review Board of the Amsterdam University Medical Centers, confirmed that the Medical Research Involving Humans Subjects Acts (WMO) did not apply, waiving the need for official approval (W18_024#18.035). Survey participation was voluntary and consent was implied through return of survey.

## 3. Results

### 3.1. Participants and Responses

All 85 ICUs in the Netherlands expressed interest in participation of whom 72/85 (85%) provided survey responses. Individuals responding on behalf of their ICUs were most commonly nurses (66/72, 92%); of whom 35/72 (49%) were advanced ventilation nurse specialists (Table 1). All ICUs were mixed medical/surgical, and both academic and non-academic hospitals were represented in the survey responses. 

### 3.2. Airway Care Practices

Airway care intervention combinations used in each ICU are displayed as a heatmap (Figure 1). We found substantial heterogeneity across ICUs in intervention combinations and in the intensity of their use (i.e., routine, as indicated or never). 

### 3.3. Heated Humidification 

Most ICUs (58/72, 81%) reported prophylactic use of heated humidification as a routine intervention in all invasively ventilated patients. A minority (11/72, 15%) used heated humidification as an ‘on indication’ treatment, with the indication defined as presence of viscous mucus. Few (3/72, 4%) ICUs never used heated humidification. 

### 3.4. Nebulization Therapy

All responding ICUs reported using nebulization of bronchodilators and/or mucolytics. In 43% (31/72) this was as a routine prophylactic intervention for bronchospasm and mucus retention with 74% (23/31) reporting routine prophylactic nebulization therapy 4 times daily. Nine (29%) of these 31 ICUs reported more frequent administration. When used “on indication”, bronchospasm or audible wheeze were the most commonly reported indications. Bronchodilators were the most commonly used drug class, independent of intensity of use. Metered dose inhalers (MDI) (37/72, 51%) or jet nebulizers (40/72, 56%) were most frequently used for nebulization therapy. Details on indications, contraindications and medication used are provided in Table 2 and Figure 1.

Although nebulization therapy was used in all responding ICUs, perception as to efficacy (17%) or necessity (28%) of prophylactic nebulization was low. Those ICUs using nebulization as a routine prophylactic intervention perceived efficacy to be higher than respondents from ICUs that used nebulization only on clinical indication (Figure 2).

### 3.5. Manual Hyperinflation

Most responding ICUs (58/72, 81%) reported using manual hyperinflation; most commonly (53/72, 74%) as on indication only. Those ICUs identified indications to include difficult oxygenation, presumed mucus presence, and decreased tidal volume. Unstable hemodynamics and active pneumothorax were the most important contraindications. Ten ICUs (10/58, 17%) reported to have no contraindications to manual hyperinflation. 

Most ICUs reported using a Mapleson CTM circuit (41/58, 71%) for manual hyperinflation. Forty-three ICUs (74%) indicated an expiration valve was used to adjust PEEP. Twenty-five ICUs (43%) ICUs using manual hyperinflation stated a predefined PEEP target was set. Few (10/58, 17%) ICUs reported using a manometer in the circuit to measure and control for high peak airway pressures. Details on indications, contraindications and materials used for manual hyperinflation are provided in Table 2 and Figure 1. Most respondents disagreed with the statement manual hyperinflation to be a safe (74%) or effective (64%) airway care intervention in invasively ventilated patients, independent of local use. (Figure 2). 

### 3.6. Mechanical Insufflation-Exsufflation

Few (16/72, 21%) ICUs reported using MI-E, with use only in response to a clinical indication such as insufficient cough strength (16/16,100%) or use of MI-E at home (10/16, 63%). MI-E was applied 2 to 3 times daily or more depending on clinical indication. Intensivists were the primary MI-E prescriber (14/16, 88%) but MI-E was applied by all ICU team members; mostly ICU nurses (15/16, 94%) and advanced ventilation nurse specialists (8/16, 50%). Years of MI-E use in the ICU setting ranged from very recent (<1 year) (3/16, 19% ICUs), 1–5 years (9/16, 56% ICUs), and 6–10 years (4/16, 25% ICUs). The majority of respondents disagreed that MI-E is a safe (75%) or effective (75%) intervention in all invasive ventilated patients, independent of local use. (Figure 2). Details on reported MI-E practices are provided in Table 2 and Figure 1.

### 3.7. Training and Education 

Most respondents described having a local protocol for nebulization therapy (57/72, 79%), manual hyperinflation (35/58, 60%) and MI-E (12/16, 75%). In 25 of the 59 (46%) ICUs that used manual hyperinflation, nurses received annual training from an expert colleague. Bedside training was the most frequently employed education method for manual hyperinflation (35/58, 60%) and MI-E (12/16, 75%).

In the post hoc analysis no differences were found in use of airway care interventions between type of hospitals or size of ICU (Appendix A). Both types of hospitals, academic-teaching and general hospitals were associated with routine use of heated humidification. The size of the ICU was not associated with its use. There were no associations of hospital type or ICU-size regarding the use of nebulization therapy. Both general and academic-teaching hospitals showed a positive association with manual hyperinflation use and ICUs > 20 beds were associated with more manual hyperinflation use. Both hospital types were associated with low use of MI-E. There was no association with size of ICU regarding MI-E (Appendix A).

## 4. Discussion

This is the first survey describing current practice of four airway care interventions within the domain of intensive care nursing for adult patients admitted to an ICU in the Netherlands. The main findings of this survey indicate substantial heterogeneity regarding the combination of airway care interventions and their intensity of use, regardless of hospital type or ICU size. This means the type of airway care received by patients depends on where in the Netherlands they are admitted.

This survey reports a high proportion of ICUs using heated humidification as routine prophylactic therapy for all ventilated patients. This is in line with previous studies in other countries [6,8,22]. However, heated humidification may not only increase workload and cost [23], but may also not be more effective compared to heat and moisture exchangers (HME) in prevention of complications. A 2017 Cochrane systematic review suggests no difference in the incidence of artificial airway occlusion, pneumonia or mortality comparing heated humidification to HME in adults and children [13]. A second systematic review in critically ill adults only confirms these findings [24]. Our data suggest knowledge translation work is needed in the Netherlands to change airway care practice from routine prophylactic use of heated humidification in all ventilated patients to use of HMEs.

Use of routine prophylactic nebulization therapy in all ventilated patients was reported by 43% of responding ICUs. Again, evidence to support this practice is limited. This practice also increases costs and nursing workload. One multi-centre randomized controlled trial comparing routine nebulization of mucolytics and bronchodilators with nebulization only on indication, showed no difference in the number of days alive and ventilator free [14]. In addition, medication side effects such as agitation occurred more frequently with prophylactic nebulizer use [14]. 

Our results show that manual hyperinflation is commonly used in the Netherlands, both as a routine prophylactic intervention or as indicated. However, the number of ICUs reporting its use has declined since a previous survey in 2009 [9]. Although alveolar recruitment and mobilization of airway secretions are cited as benefits of manual hyperinflation [10], efficacy as a routine prophylactic intervention in all invasively ventilated patients is not confirmed by evidence [25]. Furthermore, manual hyperinflation is a difficult technique to perform and, as such, may potentially harm the patient [26]. Concerns about safety of manual hyperinflation were reflected by our survey respondents. 

We found use of MI-E in invasively ventilated patients uncommon in the Netherlands (21%) compared to Canada (64% of the ICUs) [17] and the United Kingdom (98% of the ICUs) [27]. These surveys report MI-E to be used for invasively ventilated patients during weaning from invasive ventilation [17,27]. There appears to be increasing adoption of MI-E for invasively ventilated patients outside the Netherlands possibly due to the need for a safe and effective way to mobilize mucus from the lower airways. However, further research is needed as to the efficacy of MI-E in invasively ventilated critically ill patients [12]. 

### Strength and Limitations

Strength of our study is the excellent response rate meaning our data are highly generalizable to the current practice of airway care interventions of ICUs in the Netherlands. Our response rate can be attributed to following survey conduct recommendations [19], including contact by telephone prior to the survey distribution, and identification of a key respondent. Study limitations pertain to the use of a web-based self-report survey. First, by having one individual report on the practice of an ICU, responses may be reflective of perceived versus actual practice or relate to the individual’s practice rather than that of the ICU. Second, the questionnaire was designed using previous reports of airway care interventions with a focus on the domain of intensive care nursing [9,17]. Third, since we only included respondents from the Netherlands and focused on the interventions within the domain of intensive care nursing, we cannot report on other interventions applied by other health care professionals, e.g., physiotherapists. The organization of care within the intensive care differs between countries and therefore our results may be not generalizable to other countries.

## 5. Conclusions

Our survey indicates that in the Netherlands, use of prophylactic airway care interventions for heated humidification and nebulization in all invasively ventilated patients is common despite evidence of no benefit. Manual hyperinflation is frequently used, while only a minority of ICUs report using MI-E. Substantial heterogeneity exists with regard to the combination of airway care interventions and their intensity of use. The current lack of evidence and guidelines in airway care may be a reason for the heterogeneous practices we report. There is an urgent need for evidence of the benefit of these interventions, particularly when used as a routine prophylactic intervention, to inform evidence based guidelines.

## Figures and Tables

**Figure 1 jcm-10-03381-f001:**
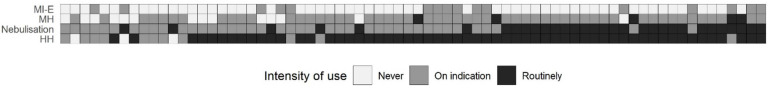
Heatmap airway care interventions in Dutch ICUs (N = 72). Current practice of heated humidification (HH), nebulization therapy, manual hyperinflation (MH) and MI-E in Dutch ICUs is graphically displayed in a heatmap. Each vertical bar is one ICU. Intensity of use of the airway care interventions is visualized by different shadings of grey according to the legend.

**Figure 2 jcm-10-03381-f002:**
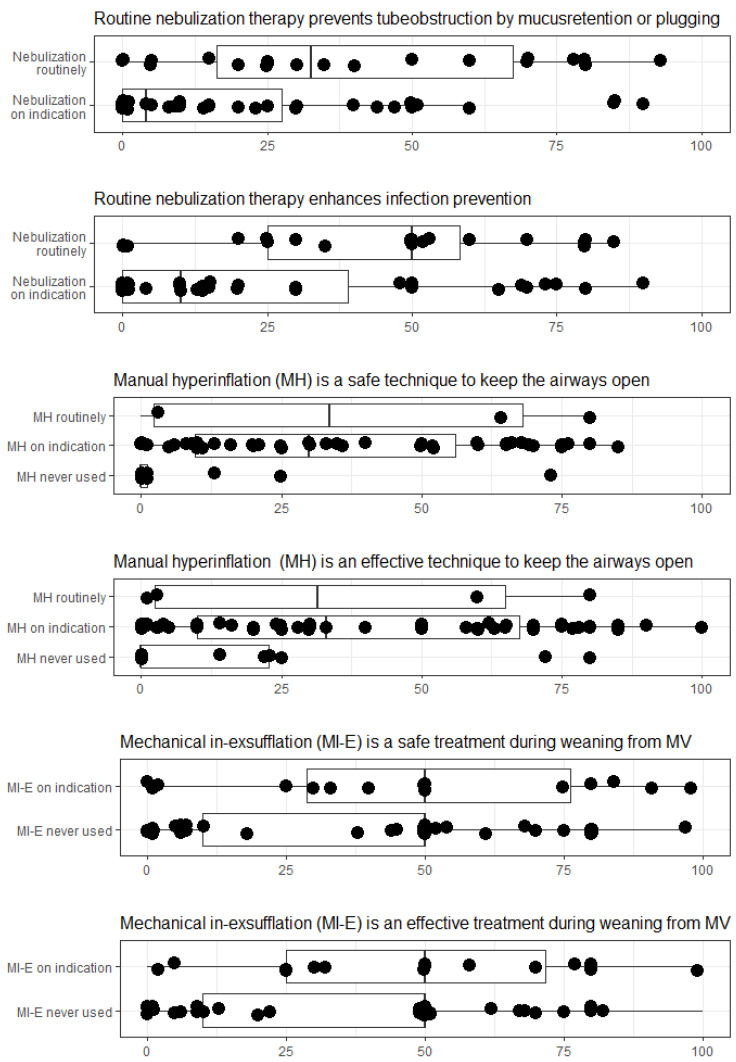
Perceptions of efficacy and safety of airway care interventions. Respondents could rate their perceptions on safety and efficacy of airway care interventions on a visual analogue scale from 0 (totally disagree) to 100 (totally agree). Results are grouped by reported intensity of use in their ICU.

**Table 1 jcm-10-03381-t001:** Demographic characteristics of respondents and Dutch ICUs (N = 72).

Characteristics	n (%)
Respondent	
ICU nurse	31 (43)
Advanced ventilation nurse specialist *	35 (49)
Intensivist	6 (8)

Hospital type	
Academic	6 (8)
Teaching ^†^	32 (44)
General	34 (47)
ICU beds available for invasive ventilation	
3–5	10 (14)
6–10	16 (22)
11–20	20 (28)
21–30	21 (29)
>30	5 (7)

* ICU nurses with additional education 14 month program on mechanical ventilation 240 study hours; ^†^ a non-academic hospital in which healthcare professionals are trained and educated. ICU, Intensive Care Unit.

**Table 2 jcm-10-03381-t002:** Airway care interventions.

Characteristics		n (%)	
**Nebulization therapy**			
Practice of use	N = 72		
routine use		31 (43)	
as treatment on indication		41 (57)	
never used		0	
Indications for use *		Bronchodilators	Mucolytics
Bronchospasm		39 (54)	4 (6)
Wheezing		37 (51)	2 (3)
in use prior to admission		29 (40)	7 (10)
decrease in tidal volume		10 (14)	2 (3)
tenacious mucus		10 (14)	33 (46)
purulent mucus		10 (14)	8 (11)
increase in peak inspiratory pressure		9 (13)	1 (1)
mucus retention		9 (13)	21 (29)
Contra-indications ***			
known drug allergy		54 (75)	
Arrhythmias		23 (32)	
pulmonary edema		5 (1)	
>15 cm H_2_O PEEP		4 (1)	
Nebulizer type ***			
jet nebulizer		40 (56)	
metered dose inhalers		38 (53)	
vibrating mesh nebulizer		22 (31)	
ultrasonic nebulizer		12 (17)	
**Manual Hyperinflation**			
Practice of use	N = 72		
routine use		5 (7)	
as treatment on indication		53 (74)	
never used		16 (19)	
Indications ***	N = 58		
difficult oxygenation		44 (76)	
presumed mucous presence		35 (60)	
decrease in tidal volume		28 (48)	
rising inspiratory pressures		24 (41)	
Contraindications ***			
unstable hemodynamics		35 (60)	
active pneumothorax		33 (57)	
intracranial hypertension		32 (55)	
>15 cm H_2_O PEEP		25 (43)	
bronchospasm		13 (22)	
pulmonary oedema		12 (20)	
Materials used			
Mapleson C© (waterset) circuit		41 (71)	
Laerdal AMBU© bag		10 (17)	
Jackson Rees-system©		1 (2)	
other ^†^		6 (10)	
**Mechanical Insufflation-Exsufflation**			
Practice of use	N = 72		
routine use		0	
as treatment on indication		16 (22)	
never used		56 (78)	
Indications *	N = 16		
insufficient cough strength		16 (100)	
already using at home		10 (63)	
repeated atelectasis		8 (50)	
regular airway care ineffectivein removing mucus		6 (38)	
prevention of reintubation		5 (31)	
prevention of intubation		4 (25)	
difficult weaning		3 (19)	
as a weaning adjunctduring all weaning		1 (6)	
prevention of pneumonia		1 (6)	
Contraindications *			
bullous emphysema		10 (63)	
severe COPD/asthma		5 (31)	
haemoptysis		6 (38)	
intracranial hypertension		9 (56)	
Device used			
Cough assist (Respironics (Philips)©		16 (100)	
Other: IPV		3 (19)	

* respondents were requested to tick all options that apply. † Mercury Medical or a combination of AMBU bag and Mapleson C waterset. Abbreviations: MV, mechanical ventilation; PEEP, Positive End Expiratory Pressure; IPV, Intra. Pulmonary Ventilation; ICU Intensive Care Unit.

## Data Availability

The survey is in Dutch. Data are available on request with the corresponding author.

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
