# Peer review of "Airway Care Interventions for Invasively Ventilated Critically Ill Adults—A Dutch National Survey"

_jcm, 2021, doi:10.3390/jcm10153381_

Round 1

Reviewer 1 Report

This is an interesting paper, which raises important questions about the need for further research into airway hygiene delivered by nurses in ICU. The paper is well written (minor grammatical errors only), and the findings of the survey are well presented. The authors should be congratulated on their high response rate, which suggests that the findings are representative of ICUs nationally. As always, there is heterogeneity between units. I am particularly interested in the relationship between nurses and physios in your country, and whether the physio undertakes some of the airway clearance treatments in those units that use manual hyperinflation (for instance) less frequently. For an international audience it may be worth, in the introduction or the discussion, stating how the physios and nurses work together, since this varies greatly between countries. I enjoyed viewing the survey figures - especially the heat map - an original way of displaying findings that worked well here. I do not have specific recommendations for improving the paper further - it is a well constructed study that is written up to a high standard. Minor revision relates to proofreading to amend small grammatical/typographical errors only.

Author Response

We thank you for giving us the opportunity to submit a revised version of our manuscript. A point-to-point reply to the comments has been described in the attached word document. 

We look forward to your response.

Kind regards, Willemke Stilma

Reviewer 2 Report

In this paper Stilma and colleagues have performed a well-organized cross-sectional survey to study some common knowledge, attitude and practices used for vented patients. Their study methods is well described and is optimal for such a study. It was interesting to observe routine use of nebulization, manual hyperinflation and mechanical in-exsufflation in ICUs. The points they make in discussion is valid and well organized. They have acknowledged main limitations of the survey as it was filled out by single person. There was also lack of response from intensivists as they would be the one to order such therapies unless they have a ventilator protocol in those places which would automatically order these.

While the paper is complete in its own way, it can be made more informative if authors can answer following.

  1. Did these places have a ventilator protocol where airway care interventions are automatically placed. If yes, how old are these. This data may not be available in the current questionnaire but would be good question to answer in subsequent study. This would help formulate interventions to replace these old protocols which sometimes persist.
  2. I would have liked to see if there was a difference in these interventions as per academic/teaching vs general hospitals to see whether this pattern was present in general hospitals. This would be easy analysis to place in supplementary appendix.
  3. Also it would be interesting to see whether size of the ICUs were related to these interventions. Comparing those with > 20 beds vs < 20 beds would be useful in discerning some aspects of those interventions.
  4. Also to see if there was personal/lack of training, comparison could be done between ICU nurse and advanced vent nurse.

Using simple logistic regression with these 3 variables to few of the outcomes such as nebs for routine use, manual hyperinflation as treatment for indication, mechanical insufflation-exsufflation as treatment on indication can help discern if ICU size, teaching status or person filling up the survey had any impact on the answers.

Author Response

Dear reviewer,

We thank you for giving us the opportunity to submit a revised version of our manuscript. A point-to-point reply to the comments has been described in the attached word document. 

We look forward to your response.

Kind regards, Willemke Stilma

Reviewer 3 Report

The authors describe the current practices and perception on routine use of various strategies for airway clearance in critically ill patients who undergo invasive mechanical ventilation. I must congratulate the authors for conducting this very important survey and highlighting a crucial topic that is relevant to daily care of critically ill patients in the ICU. The study highlights an important knowledge gap in airway clearance methods and heterogeneity in methods used in different institutions. I believe this is a phenomenal manuscript that is well-written, succint and provides continuous interest to the directed readership. 

Minor recommendations:

  1. Did the authors mean "Bronchodilators" in table 2 instead of "Broncholytics" in Indication of use section? If so, please correct
  2.  In figure 2, the first plot has a the sentence with all words clumped together without a space in between, would recommend correcting it.
  3. I would recommend the authors to switch the term "sputum" to airway secretions as sputum is considered more upper airway secretions compared the mucoid secretions produced in lower airways. A common and neutral term would be "airway secretions" and "airway clearance measures".

Congratulations on the phenomenal manuscript. 

Author Response

(The authors gave the same response as above.)

Round 2

Reviewer 1 Report

Thank you for addressing the comment regarding physio/nurses. This paper will be of interest to a wide audience.

Reviewer 2 Report

Thank you for the additions of supplementary tables.

I have no further questions.

Reviewer 3 Report

This reviewer commends the authors for making all the changes that this reviewer had suggested. The manuscript is brilliant and this reviewer recommend it for publication